# The Impact of Government Governance on Regional Public Health and Development Measures from the Perspective of Ecological Environment

**DOI:** 10.3390/ijerph20043286

**Published:** 2023-02-13

**Authors:** Tao He, Lulu Liu, Manyi Gu

**Affiliations:** 1School of Political Science & International Relations, Tongji University, Shanghai 200092, China; 2Party Committee Propaganda Department & United Front Work Department, Southwest Medical University, Luzhou 646000, China; 3School of Humanities and Management, Southwest Medical University, Luzhou 646000, China

**Keywords:** government governance, public health, ecological environment

## Abstract

In order to further improve the satisfaction of public health safety, this paper discusses the impact of government governance on the satisfaction of regional public health safety and discusses the effectiveness of government public health governance and development countermeasures. From the perspective of ecological environment protection, combined with the survey data of national urban public health safety satisfaction in the last two years, this paper performs an in-depth empirical analysis on the relationship between government governance, public health governance efficiency, public credibility and regional public health safety satisfaction, as well as the impact mechanism. Through the analysis, it is found that the efficiency of government governance directly affects the satisfaction of regional residents with public health safety. With the help of the intermediary effect test, the significant level standard error of the indirect effect is greater than 1.96, and the confidence interval does not include 0, which proves that the intermediary effect exists. On this basis, the strategy of improving the satisfaction of regional public health security is further analyzed.

## 1. Introduction

In recent years, with the gradual deterioration of the global ecological environment, eco-environmental protection has become a hot topic of general concern all over the world. Affected by this, the improvement of the quality of regional public health has also received general attention from all walks of life. The government plays an important role in regional public health governance [1].

With the implementation and continuous expansion of China’s basic public health services in 2009, scholars have successively investigated the awareness rate and satisfaction of residents’ public health services around the country. Many research results show that Chinese residents’ satisfaction with public health services shows the characteristics of uneven geographical distribution [2]. Although many achievements have been made in the construction of Chinese government efficiency, relatively speaking, empirical research on the factors affected by government efficiency is not sufficient, and most of these factors focus on the impact of regional and even national macro policy implementation, including talent gathering, innovation and entrepreneurship, business environment and other fields.

Some scholars have carried out research on the impact and role of administrative efficiency on the willingness of high-level talent agglomeration, and through empirical testing, it is found that administrative efficiency, as one of the dimensions of administrative efficiency, has a significant positive effect on high-level talent agglomeration. Government regulation mainly includes economic regulation and social regulation. Economic regulation is mainly aimed at the problem of information asymmetry in reality, preventing the irrational allocation of resources, which can ensure the fair use of demanders. Social regulation is based on the perspective of national security and public interests to implement mandatory regulation on the social environment, natural resources, security and other aspects. The influence of the government on the demand of urban community health service mainly lies in the integration of market regulation and community management through policy guidance, financial support, strengthening publicity and other strategies and methods; following the basic laws of the medical service market; doing a good job of “distribution according to demand”; and transforming the potential health service demand of urban community residents into the actual distribution demand. This is in order to maximize the utilization of urban community health service resources to better meet the needs of both supply and demand, so as to realize their macro-control role and perform their regulatory functions.

Based on the global data from 1995 to 2022, some scholars found that government efficiency has a significant positive effect on national innovation capacity. In terms of exploring the impact of government effectiveness on the individual level, the relevant empirical research is less. Among them, some scholars have systematically studied the impact and mechanism of government effectiveness on residents’ well-being. Based on the practical needs of the development of the Zhuhai government and the theory of social exchange, they used a structural equation model to discuss in detail the mechanism of a city’s government effectiveness on residents’ well-being. Thus, it reveals the positive impact of government effectiveness on residents’ well-being and further clarifies the regulatory role of urban belonging in the impact of government effectiveness on residents’ well-being, as well as the intermediary role of residents’ living conditions in this impact process [3]. In order to achieve the fragile ecological environment, it is necessary to accurately identify the key environmental factors affecting the economic development of each province, giving full play to the role of government governance measures, and realize the benign economic development of each province.

## 2. Empirical Research

The relationship between the main variables in this study is shown in Figure 1:

### 2.1. Research Data Sources

The data used in this study are from the 2017 national urban public security survey data in the urban public security database of the China University of Mining and Technology. In terms of the demographic characteristics of the respondents, statistics show that among all the residents who completed the questionnaire (excluding missing values), men accounted for 50.0% and women accounted for 50.0% [4]. After removing the missing values, the basic information of all residents who filled in the questionnaire in this data survey is shown in Table 1.

### 2.2. Variable Measurement

#### 2.2.1. Exploratory Factor Analysis

Before conducting the evaluation of the index, it was necessary to determine whether the index could be used for the analysis of the index itself. This can be checked by two key indicators: the KMO value of sampling quality and the result of Bartlett’s test of sphericity [5]. In this study, KMO and Bartlett’s tests were performed on different measures of public health satisfaction, and the results of the analysis are shown in Table 2.

The final result of the factor analysis is shown in Table 3. The cumulative contribution rate of the two components is 73.558%, that is, the cumulative interpretation of the total variance is 73.558%, which is greater than the 0.6 suggested by hair. The observation variables xn1, xn2, xn3 and xn4 have a high load on factor 1. As these observation variables mainly reflect the subjective perception of the surveyed urban residents on the effectiveness of government public health governance, factor 1 is named as the effectiveness factor of government public health governance [6].

In addition, the internal quality of each aircraft model can also be checked. The internal quality of the facet can be measured and verified by the reliability and mean difference of the measured values. According to Table 4, the average difference between the two different abilities is 0.696 and 0.558: both of them are higher than the standard 0.50, and the reliability is 0.901 and 0.790. A value of 0.60 indicates that the quality of the model is reasonable [7].

#### 2.2.2. Normality Test of Sample Data

In this paper, spss26.0 was used to test the normality of each variable in the data model. The results of the clinical trials are shown in Table 5.

It can be seen that the skewness coefficients and kurtosis coefficients of all observed variables in the sample data meet the test criteria, indicating that the data conform to the characteristics of normal distribution. Therefore, we can use the structural equation model to further analyze and verify the research hypothesis [8].

## 3. Current Situation of Regional Public Health Supply Structure under Government Governance

### 3.1. Main Structure of Public Health Supply

It can be seen from the data in Figure 2 below that the health spending of the Chinese government has not changed significantly in recent years. The distribution of government health spending has decreased from 2018 to 2021, from 30.7% in 2018; the lowest share of health spending in 2021 was 29.96%, and in 2022 increased to 30.45%, but remained lower than the government’s 2018 medical expenses.

### 3.2. Public Health Expenditure Structure

From the perspective of China’s public health care, Table 6 shows that China’s health care spending has increased year by year, from CNY 132.02 billion in 2013 to CNY 119.531 billion in 2018. In 2022, the price pattern also shows the pattern of financing. This shows that the increase in public health spending is mostly influenced by the budget. In 2013, 1.9% of medical and health care costs came from the central government, and 98.1% was public expenditure; meanwhile, in 2022, the central government accounted for only 0.7% of medical and health spending, while public spending accounted for 99.3% [9].

### 3.3. Supply Structure of Public Health in the East, Middle and West

From the point of view of the region, the per capita health in the eastern region is higher than other regions, which is the reason for the regional disparity differences in population health in the eastern region. Spending on health per person in the eastern region is lower than other regions, which affects per capita health spending; in other regions in the east, there is little difference in the scale of health spending per capita. The difference in the index of public health expenditure per capita between the regions is small and insignificant; the level of health expenditure per person in the eastern region is higher than that of the middle region, and the level of health of the population in the eastern region is higher than on the middle ground [10].

To identify regional differences in China’s health spending, the Theil index in the eastern region gradually decreased from 2018 to 2020, as can be seen from Theil index line in Figure 3 and the price changes in Table 7. It shows that the difference in health care costs of the population in the east is gradually decreasing. From 2018 to 2022, the TAIR index of the eastern region is the largest, and the power line of the TAIR index is at the top.

## 4. Empirical Analysis of the Impact of Government Public Health Governance Efficiency and Government Credibility on Public Health Safety Satisfaction

### 4.1. Descriptive Statistical Analysis

#### 4.1.1. Public Health Safety Satisfaction Profile

The overall evaluation of urban residents’ satisfaction with public health and safety is shown in Table 8. In general, the average score of the public health safety evaluation of the interviewed residents is 3.350, which is above the medium level [11]. However, under the macro social background of the increasing living standards of Chinese residents and the increasing needs and expectations of public safety, it is still questionable whether the above medium public health safety evaluation indicates that the public health safety situation has been widely recognized by urban residents.

#### 4.1.2. Overview of Government Public Health Governance Effectiveness

The overall situation of urban residents’ perception of government public health governance efficiency is shown in Table 9. Overall, the average score of the overall effectiveness of public health governance is 5.059, which is only at the medium level. The respondents scored the highest on the sub item of emergency management efficiency 5.191; the average score of normative effectiveness was the lowest at 4.919 [12]. This shows that the effectiveness of government public health governance at the subjective perception level of urban residents has great room for improvement, both on the whole and in specific sub items.

#### 4.1.3. Overview of Government Credibility

The overall perception of urban residents on the evaluation of government credibility is shown in Table 10. In general, the average government credibility is 3.280, less than 3.5. Although the level of government credibility is above the medium level, it is not blindly optimistic, and there is still a risk of a weak foundation. The average scores of the surveyed residents on the degree of realization of the government’s expectations, confidence and trust were 3.191, 3.297 and 3.351, respectively. The specific sub items of government credibility and the evaluation of public health safety satisfaction show the same distribution characteristics of large in the middle and small at both ends [13].

#### 4.1.4. Correlation Analysis between Variables

By averaging the measurement items of government public health governance efficiency and government credibility, respectively, the correlation between the three main research variables can be obtained. The results are shown in Table 11. There is a positive correlation between the effectiveness of government public health governance, government credibility and urban residents’ satisfaction with public health safety at a significance level of 0.01 [14].

### 4.2. Main Effect Test

#### 4.2.1. Overview of Structural Equation Models

A structural equation model is composed of a measurement model and structural model. The measurement equation is:(1)$$X=\Lambda_{x}\xi +\delta$$
(2)$$Y=\Lambda_{y}\eta +\varepsilon$$

The structural equation is:(3)η=Bη+Γξ+ξ

Equation (1) is the measurement equation of the external latent variable, and ξ is the external latent variable; Equation (2) is the measurement equation of the internal dependent latent variable, and η is the internal dependent latent variable; Equation (3) is a structural equation, which describes the relationship between internal latent variables η and each other, and Γ describes the influence of external latent variables ς on internal latent variables η.

#### 4.2.2. Empirical Test and Results

In this study, because public health safety satisfaction is an explicit variable, while the government’s public health governance efficiency and government credibility are regarded as potential variables, and the structural model of path analysis includes both explicit and potential variables, the path analysis of the mixed model is constructed. The overall model fitness test results of the overall relationship model are shown in Table 12.

It can be seen that, except for the x^2^/df value and PGFI value, the other statistical test quantities meet the overall model fitness standard. As mentioned in Section 3, the problem of too large an x^2^/df value caused by too large a sample size is also ignored here. Therefore, the fitness index of the model generally meets the fitness standard, indicating that the model has a good fitness. Table 13 shows the test results of the overall relationship model [15].

Table 14 shows the hypothesis test results. It can be seen that the standardized regression coefficients of “government public health governance efficiency” on “government credibility” and “public health safety satisfaction” are 0.232 and 0.251, respectively, and the standardized regression coefficients of “government credibility” on “public health safety satisfaction” are 0.326, both reaching a significant level of 0.001. The value of standardized regression coefficient is positive, indicating that the influence path relationship between the three core variables is positive, which is consistent with the original theoretical hypothesis [16].

This means that the first three hypotheses of this study have been verified. Specifically:

First, the operation of the government’s public health care has a positive effect on the satisfaction of the public’s health security. The sign of the path coefficient of the influence of public health management on the results of urban residents’ satisfaction with public health safety is positive and reaches a significant level of 0.001, indicating that the effectiveness of public health management has a positive effect on the assessment of health. In terms of the satisfaction of residents in cities with public health safety, hypothesis H1 is confirmed [17].

Second, the effectiveness of government public health care has a positive effect on trust in government. The sign of the path coefficient of the effect of health management on public trust is positive and reaches a significant level of 0.001, indicating that public health management has a positive effect on public confidence according to citizens, meaning that hypothesis H2 is confirmed.

### 4.3. Intermediary Effect Test

#### 4.3.1. Overview of Mediation Effect

Mediation means that there are m intervening variables in the causal relationship mechanism between the independent variable *X* and the dependent variable *Y*: that is, the independent variable *X* can affect the difference between *Y* and M mediating variables [18]. In this case, m is the average difference between m affecting both the independent variable *X* and the variable *Y*. A schematic diagram of the mediation process is shown in Figure 4.

When all variables are centralized or standardized, the specific relationship between the three variables can be described by the following equation:(4)$$Y=cX+e_{1}$$
(5)$$M=aX+e_{2}$$
(6)$$Y=c^{\prime }X+bM+e_{3}$$

The intermediary effect is equal to the coefficient product ab, that is, the indirect effect. The total effect is the sum of the direct effect and the indirect effect. The specific relationship is as follows:(7)$$c=c^{\prime }+ab$$

The mediating effect can be divided into the complete mediating effect and the partial mediating effect. In the complete intermediary effect, *X* can only affect *Y* through the intermediary variable m, but it cannot directly affect *Y*: that is, at this time, the coefficient $c^{\prime }$ is 0, *c* = *a**b*. In some mediating effects, *X* can not only directly affect *Y*, but also affect *Y* through the mediating variable M.

#### 4.3.2. Intermediary Effect Test

The bootstrap method is one of the common methods to test the mediation effect. It does not need samples to obey the hypothesis of positive distribution and does not rely on theoretical standard error. It has high statistical effect, and is also considered the most ideal mediation effect test method at present. Bootstrap is a nonparametric statistical method that estimates the variance of statistics and then estimates the interval. Its basic core idea is to use resampled sample data to calculate statistics and estimate sample distribution [19]. The test results are shown in Table 15.

Table 15 shows that government trust plays a mediating role in the effect of government health care on public health on urban residents’ satisfaction with public services. The health Z value of the significance level of the indirect effect is equal to the value of the unstandardized coefficient of the indirect effect/standard error of the indirect effect = 0.034/0.002 > 1.96. In addition, the confidence interval for the direct effect does not include 0, thus confirming the existence of a moderated effect. The planting time for the direct effect does not include 0, so there is still an indirect effect [20]. Through the analysis, it is found that the efficiency of government governance directly affects the satisfaction of regional residents with public health security. With the help of the mediation effect test, the significant level standard error of the indirect effect is greater than 1.96, and the confidence interval does not include 0, demonstrating the existence of the mediation effect. On this basis, strategies to improve regional public health safety satisfaction were further analyzed.

## 5. Development Countermeasures of Government Governance to Improve the Satisfaction and Quality of Public Health Security from the Perspective of Ecological Environment

### 5.1. Improve the Existing Public Health Administrative Norms

First, governance plays an important role in improving the legal system of public health, establishing and forming the legal department of public health, improving the status of public health law in China’s legal system, and making public health law an independent department of the legal system. Second, it solves the basic theoretical problems related to public health law (characteristics, essence, origin, legal basic rights and obligations of health) and determines the basic categories of public health law to adjust various social relations based on the right to life and health [21]. This is to explore the relationship between public health law and other legal norms, and to study the relationship between public health legal rights and obligations, and the related legal responsibilities, so as to provide a theoretical basis for the establishment of the basic theoretical system of health law. Third, it is of great significance to strengthen the construction of a socialist public health legal system. (1) In terms of public health legislation, through the study of the legal system of public health, we can clearly find the defects and deficiencies of the current legal system of public health, so as to scientifically formulate public health legislation strategies; gradually implement public health legislation planning; realize legislative intent, legislative value and legislative effectiveness; and provide theoretical support for the construction of public health rule of law. (2) It is of great significance to guide public health administrative law enforcement and judicial practice, improve the health level and awareness of the safeguarding rights of the whole population, and protect citizens’ health rights and interests. Only by enhancing the understanding and recognition of the national public health legal system can the public increase their knowledge, understanding of, and ability to abide by the law [22].

### 5.2. Formulate the Basic Law of Public Health and Standardize Government Functions

The formal rationality of law can serve as a powerful footnote to the systematization requirements of public health law. The so-called formal rationality of law refers to the systematization and scientization of legal rules controlled by reason and the formalization of the process of law making and application. According to Max Weber, a famous German legal sociologist, formal rationality is the highest level of rationality pursued by law. He believes that theoretically, the last stage of legal development is the systematic development of professional jurists on the basis of literature and formal logic training [23]. However, due to the lack of command and coordination, separate public health laws and regulations are isolated from each other. This situation is not conducive to the unified regulation and management of the field of public health, nor to the implementation of a single public health law. People involved in legislation at different places often start from the perspective of their own departments and regions, causing the relevant laws and regulations to be one-sided and limited, rather than overall and long term. Therefore, it is necessary to formulate a unified basic law of public health.

### 5.3. Give Full Play to the Government’s Supervision of Public Health Services

The government’s supervision function in the field of public health is particularly important under the current market economy. Since the reform and opening up, under the conditions of the market economy, due to the profit seeking nature of the market, many bad businesses, in order to pursue interests, have taken advantage of problems such as untimely government supervision and loopholes in supervision, waiting for the opportunity to produce and sell fake and shoddy food and drugs, causing serious social harm, as well as causing public opinion to falter regarding the government’s supervision ability. China has had a market economy for more than 30 years [24]. With the rapid economic development, weak supervision of market behavior has occurred from time to time, especially in the field of public health. The field of public health is closely related to all aspects of society; whether it is food safety or drug safety, environmental protection or ecological balance, it is related to the lives and safety of the public. It is also the strictest set of regulatory measures that public health functional departments must implement. From production license, production to transportation, sales and other links, professional supervisors must be sent to supervise the whole process, in order to be open and transparent and ensure that businesses can conduct market behavior in accordance with relevant professional standards [25].

## 6. Conclusions

In the new era of social transformation, various public health events emerge in an endless stream; this leads to a great threat to public health and causes people to question the government’s governance capacity in the field of public health. In this case, the government should aim at building a service-oriented government, give full play to its initiative in performing public health functions, and strive to reform and improve China’s public health undertakings under the guidance of public health law.

## Figures and Tables

**Figure 1 ijerph-20-03286-f001:**
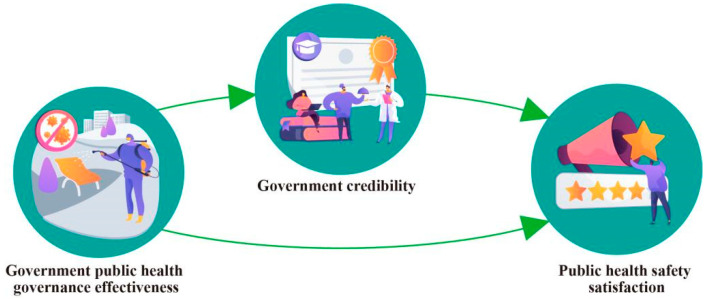
Schematic diagram of research model.

**Figure 2 ijerph-20-03286-f002:**
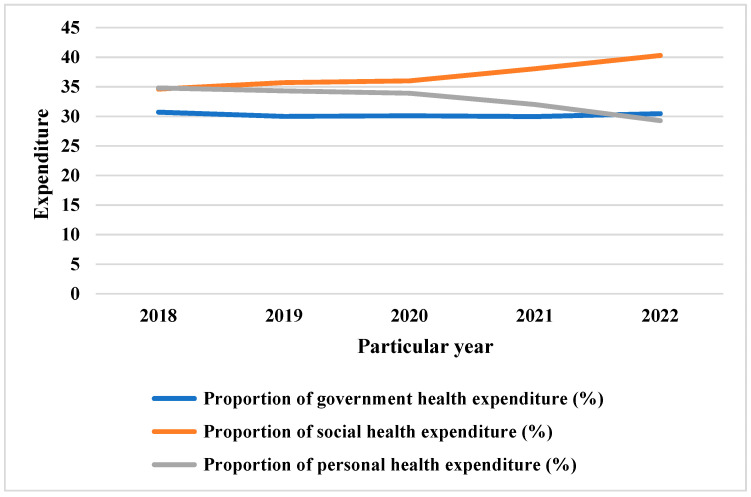
Main structure of public health expenditure in China.

**Figure 3 ijerph-20-03286-f003:**
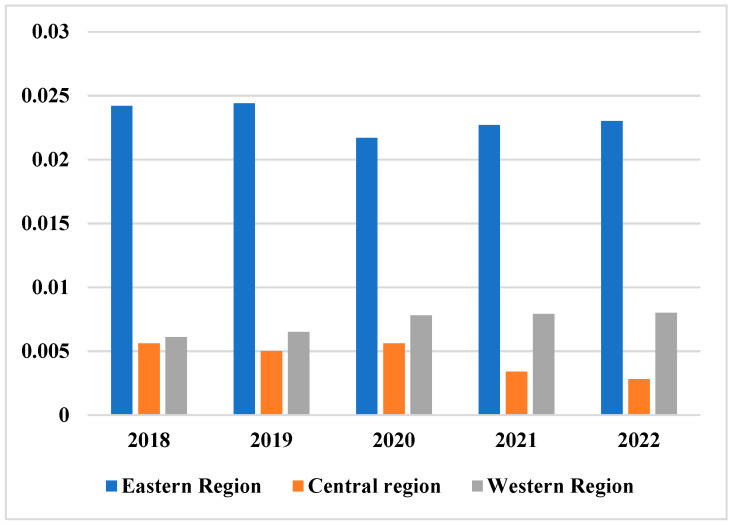
Change value of public health Theil index of fiscal expenditure in eastern, central and western regions.

**Figure 4 ijerph-20-03286-f004:**
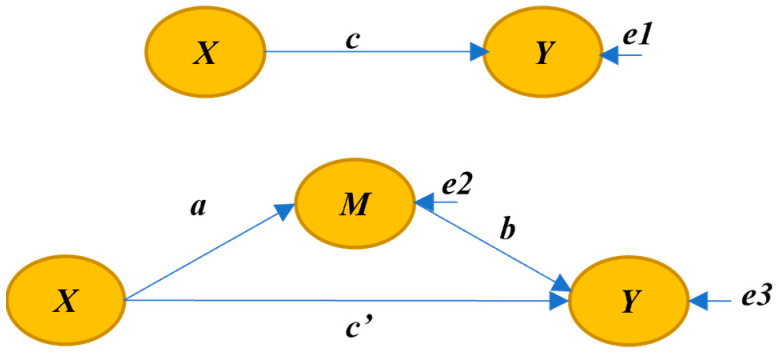
Schematic diagram of intermediary effect.

**Table 1 ijerph-20-03286-t001:** Statistical table of residents’ basic information filled in the questionnaire.

Variable	Category	Frequency	Frequency%	Variable	Category	Frequency	Frequency%
Political outlook	Member of the Communist Party of China	1668	18.0		18–29 years old	4453	48.0
	Democratic party’s	228	2.5	Age	30–44 years old	2680	28.9
	Communist Youth League member	2770	29.9		45–59 years old	1588	17.1
	Masses	4580	49.4		Over 60 years old	544	5.9
Gender	Male	4632	50.0		Civil servant	342	3.7
	Female	4639	50.0		Personnel of public institutions	1061	11.4
Degree of education	Primary school and below	342	3.7		Clerk	2031	21.9
	Junior high school	1139	12.3	Identity	Migrant workers	485	5.2
	High school (vocational and technical secondary school)	2371	25.6		Student	2414	26.0
	University (junior college)	4868	52.5		Professional	1396	15.1
	Graduate and above	546	5.9		Retired personnel	532	5.7
					Other	996	10.7
Nation	Han nationality	8119	87.6	Religious belief	Nothing	7781	83.9
	Zhuang nationality	166	1.8		Buddhism	775	8.4
	Manchu	184	2.0		Taoism	95	1.0
	Hui nationality	240	2.6		Christianity	198	2.1
	Miao nationality	63	0.7		Islamism	214	2.3
	Uygur ethnic group	56	0.6		Catholicism	38	0.4
	Tujia nationality	37	0.4		Other	157	1.7
	Yi nationality	26	0.3	Household registration type	City of this city	4816	51.9
	Mongolian	60	0.6		Rural areas of the city	1357	14.6
	The Zang or Tibetan people	207	2.2		Foreign cities	1719	18.5
	Other	113	1.2		Rural areas outside the city	1369	14.8
Personal monthly income	Below CNY 2000	2734	9.5	Personal monthly income	CNY 5001–8000	1325	14.3
	CNY 2001–3500	2109	22.7		CNY 8001–12,500	430	4.6
	CNY 3501–5000	2288	24.7		More than CNY 12,500	202	2.2

**Table 2 ijerph-20-03286-t002:** KMO and Bartlett’s test of the scale.

KMO Sampling Suitability Quantity		0.811
Bartlett’s Sphericity Test	Approximate chi square	30,531.954
	Freedom	21
	Significance	0.000

**Table 3 ijerph-20-03286-t003:** Total variation and rotated component matrix of factor interpretation extracted from the scale.

	Question Item	Ingredients	
		1	2
Government public health governance effectiveness (GX)	Prevention effectiveness (xn1)	0.835	0.106
	Regulatory effectiveness (xn2)	0.889	0.076
	Specification efficiency (xn3)	0.881	0.069
	Emergency management efficiency (xn4)	0.861	0.128
Government credibility (GX)	Government expectation (gx1)	0.102	0.812
	Government confidence (gx2)	0.090	0.859
	Government trust (gx3)	0.080	0.828
	Variance explanatory quantity	43.273%	30.285%
	Cumulative variance interpretation		73.558%

**Table 4 ijerph-20-03286-t004:** Test results of internal structure suitability of the model by confirmatory factor analysis.

Potential Variables	Observation Variables	Standardized Load	Index Reliability	Measurement Error	Composite Reliability	Mean Variance Extraction
XG	Preventive effectiveness	0.757	0.573	0.427	0.901	0.696
	Regulatory effectiveness	0.892	0.796	0.204		
	Normative efficiency	0.827	0.684	0.316		
GX	Emergency management efficiency	0.856	0.733	0.267	0.790	0.558
	Government expectations	0.696	0.487	0.513		
	Government confidence	0.814	0.663	0.337		
	Government trust	0.723	0.523	0.477		

**Table 5 ijerph-20-03286-t005:** Normality test results of the scale.

	Number of Cases	Average Value	Standard Deviation	Skewness	Kurtosis
Xn1	9273	5.10	2.674	0.147	−0.948
Xn2	9273	5.02	2.807	0.173	−1.050
Xn3	9273	4.92	2.691	0.192	−0.959
Xn4	9273	5.19	2.652	0.067	−0.959
my	9273	3.35	0.888	−0.223	−0.009
Gx1	9273	3.19	0.780	−0.283	0.611
Gx2	9273	3.30	0.817	−0.321	0.381
Gx3	9273	3.35	0.815	−0.366	0.364

**Table 6 ijerph-20-03286-t006:** Structure of government expenditure on health care in China from 2013 to 2022 unit: CNY 100 million, percentage%.

Particular Year	Total Medical and Health Expenditure	Central Expenditure	Local Expenditure	Central Proportion	Local Proportion
2018	6459	71.3	6358.2	1.1	98.9
2019	7245.1	74.3	7170.8	1	99
2020	8279.9	76.7	8203.2	0.9	99.1
2021	10,176.8	90.1	10,086.7	0.9	99.1
2022	11,953.1	84.4	11,868.7	0.7	99.3

**Table 7 ijerph-20-03286-t007:** Changes of Theil index of public health in eastern, central and western regions.

Intra Group Theil Index	2018	2019	2020	2021	2022
Eastern region	0.0242	0.0244	0.0217	0.0227	0.0230
Central region	0.0056	0.0050	0.0056	0.0034	0.0028
Western region	0.0061	0.0065	0.0078	0.0079	0.0080

**Table 8 ijerph-20-03286-t008:** Overall satisfaction with public health and safety (unit: %).

Project	Public Health Safety Evaluation Score					Mean Value
	Very Poor	Relatively Poor	Commonly	Better	Very Nice	1–5
Public health safety satisfaction	2.5	12.1	42.1	34.8	8.5	3.350

**Table 9 ijerph-20-03286-t009:** Perceived effectiveness of government public health governance (unit: %).

Project	Rating Score of Worry Degree (Extremely Worried–Not Worried at all)										Mean Value
	1	2	3	4	5	6	7	9	10	1–10	5.059
Preventive effectiveness	12.1	7.3	11.7	11.2	15.8	10.0	9.6	9.9	5.0	7.2	5.104
Regulatory effectiveness	14.9	8.1	10.6	10.8	13.8	9.6	9.2	9.2	5.3	8.2	5.024
Normative efficiency	14.5	7.7	11.6	11.7	14.6	10.6	8.7	9.4	5.1	6.2	4.919
Emergency management efficiency	11.7	7.0	10.6	11.1	15.2	11.4	10.4	10.2	6.1	6.4	5.191

**Table 10 ijerph-20-03286-t010:** Perception of government credibility (unit:%).

Project	Evaluation Score of Government Credibility					Mean Value
	1	2	3	4	5	(1–5)
Government credibility						3.280
Expected degree	2.7	11.2	53.6	29.2	3.3	3.191
Confidence level	2.5	10.4	47.0	35.1	5.0	3.297
Degree of trust	2.2	9.7	44.4	38.2	5.5	3.351

**Table 11 ijerph-20-03286-t011:** Correlation analysis between variables.

Variable	Relevance	XN	GX	MY
Government public health governance effectiveness (XN)	Pearson correlation	1		
Government credibility (GX)	Pearson correlation Sig (double tailed)	0.215 ** 0.000	1	
Public health safety satisfaction (MY)	Pearson correlation Sig (double tailed)	0.318 ** 0.000	0.345 ** 0.000	1

Note: ** *p* < 0.01.

**Table 12 ijerph-20-03286-t012:** Overall model fitness test results of overall relationship model.

Test Statistics	Adaptation Standard	Inspection Result Data	Model Adaptation Judgment
	Absolute fitness index		
GFI value	>0.90	0.997	Yes
AGFI value	>0.90	0.994	Yes
RMR value	<0.05	0.026	Yes
RMSEA value	<0.05 (good) <0.08 (reasonable)	0.023	Good
	Value added fitness index		
NFI value	>0.90	0.997	Yes
RFI value	>0.90	0.995	Yes
CFI value	>0.90	0.998	Yes
	Simple fitness index		
*X*^2^/df value	1 < x/df value < 3, good; 3 < x2/df value < 5, acceptable; 5 < x2/df value, poor	5.987	Poor
PNFI value	>0.50	0.570	Yes
PGFI value	>0.50	0.443	No

**Table 13 ijerph-20-03286-t013:** Test results of the overall relationship model.

Relationship between Variables	Non-Standardized Estimation Results				Standardized Regression Coefficient
	Non-Standardized Regression Coefficient	Standard Error	Critical Ratio Value	Significance	
xn1←XN	1				0.802
xn2←XN	1.103	0.013	83.511	……	0.844
xn3←XN	1.098	0.013	83.805	……	0.875
xn4←XN	0.999	0.012	80.252	……	0.808
gx1←GX	0.704			……	0.704
gx2←GX	1.2	0.021	58.317	……	0.807
gx3←GX	1.075	0.019	56.995	……	0.724

Note: The symbol “←” indicates the action direction between variables.

**Table 14 ijerph-20-03286-t014:** Research assumptions and test results.

Relationship Between Variables	Non-Standardized Path Coefficient	Standardized Path Coefficient	Corresponding Assumption	Hypothesis Test Results
MY←XN	0.104	0.251 ***	H1	Support
GX Y←XN	0.059	0.232 ***	H2	Support
MY Y←GX	0.527	0.326 ***	H3	Support

Note: *** *p* < 0.001; The symbol “←” indicates the action direction between variables.

**Table 15 ijerph-20-03286-t015:** Intermediary effect test results.

Effect Type	Non-Standardized Test Results		Bias-Corrected		Percentile	
	Coefficient	Standard Error	95% Confidence Interval	*p*	95% Confidence Interval	*p*
Total effect	0.143	0.005	0.133, 0.153	0.001	0.133, 0.153	0.001
Indirect effect	0.034	0.002	0.030, 0.039	0.001	0.030, 0.039	0.001
Direct effect	0.109	0.005	0100, 0.118	0.001	0.100, 0.118	0.001

## Data Availability

The labeled data set used to support that findings of this study are available from the corresponding author upon request.

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
