# Peer review of "The Impact of Government Governance on Regional Public Health and Development Measures from the Perspective of Ecological Environment"

_ijerph, 2023, doi:10.3390/ijerph20043286_

Round 1

Reviewer 1 Report

In order to further improve the satisfaction of public health safety, this paper discusses the impact of government governance on the satisfaction of regional public health safety, and discusses the effectiveness of government public health governance and development countermeasures.The article has many problems and needs major revision.

1. The reference format is not uniform, such as 18, 22, 25.

2. The reference data is relatively old, and Table 6 uses the data from 2011-2015. The data analysis using the last five years is more telling.

3. There are too many tables, which can be converted into pictures or put into supporting materials.

4. The listed data only illustrate the trend without in-depth analysis of the reasons.

Author Response

1. The reference format is not uniform, such as 18, 22, 25.

Re: Thank you for your comments. I have modified it according to your comments.

2. The reference data is relatively old, and Table 6 uses the data from 2011-2015. The data analysis using the last five years is more telling.

Re: Thank you for your comments. I have modified it according to your comments.

3. There are too many tables, which can be converted into pictures or put into supporting materials.

Re: Thank you for your comments. I have made modifications according to your comments. Re: Thank you for your comments. I have made modifications according to your comments.

4. The listed data only illustrate the trend without in-depth analysis of the reasons.

Re: Thank you for your comments. I have revised it according to your opinion: Through analysis, it is found that the efficiency of government governance directly affects regional residents' satisfaction with public health security. By means of mediating effect test, the significance level standard error of indirect effect is greater than 1.96, and the confidence interval does not include 0, which proves the existence of mediating effect. On this basis, it further analyzes the strategies to improve the satisfaction of regional public health security.

Reviewer 2 Report

MS: The impact of government governance on regional public health and development measures from the perspective of ecological environment by Liu & Gu 

From the perspective of ecological environment, this paper aimed to explore the relationship between government governance and public health governance efficiency, public credibility and regional public health safety satisfaction. The authors found that the efficiency of government governance directly affected the satisfaction of regional residents with public health safety.

Although the analyses of this study were well conducted, I still have several major concerns.

1. It remains unclear in the section of introduction why this study was necessary to conduct, and why major objectives the authors would achieve.

2. No variables related to ecological environment were involved at all in the analyses of the study. How could the authors reveal the influences of government governance on regional public health and development measures from the perspective of ecological environment?

3. The data source of this study might probably be a problem. The data used in this study were derived from the 2017 national urban public security survey data in the urban public security database of China University of Mining and Technology. However, none of the authors is from this institution, and the institution which provided the data was not acknowledged at all by the authors. How did the authors acquire these data?

4. The conclusions of this study was obtained only based on the 2017 national urban public security survey data in the urban public security database of China University of Mining and Technology. Attention should be taken that the government governance on regional public health and development measures might have changed a lot. I do not think the results of this study could reflect the current impact of government governance on regional public health.

5. In the section of conclusion, the authors stated “, and also makes the father-in-law question the government's governance ability in the field of public health”. What the hell the authors mean???

Author Response

  1. It remains unclear in the section of introduction why this study was necessary to conduct, and why major objectives the authors would achieve.

Re:Thank you for your comments. I have made modifications according to your comments:In order to achieve the fragile ecological environment, it is necessary to accurately find out the key environmental factors affecting the economic development of each province, give full play to the role of government governance measures, and realize the benign economic development of each province.

  1. No variables related to ecological environment were involved at all in the analyses of the study. How could the authors reveal the influences of government governance on regional public health and development measures from the perspective of ecological environment?

Re:Thank you for your comments. I have made modifications according to your comments:Government regulation mainly includes economic regulation and social regulation. Among them, the economic regulation is mainly aimed at the problem of information asymmetry in reality, preventing the irrational allocation of resources, which can ensure the fair use of demanders. Social regulation is based on the perspective of national security and public interests to implement mandatory regulation on social environment, natural resources, security and other aspects. The influence of the government on the demand of urban community health service mainly lies in the integration of market regulation and community management through policy guidance, financial support, strengthening publicity and other strategies and methods, following the basic laws of the medical service market, doing a good job of "distribution according to demand", transforming the potential health service demand of urban community residents into the actual distribution demand, so as to maximize the utilization of urban community health service resources, To better meet the needs of both the supply and demand sides, so as to realize their macro-control role and perform their regulatory functions.

  1. The data source of this study might probably be a problem. The data used in this study were derived from the 2017 national urban public security survey data in the urban public security database of China University of Mining and Technology. However, none of the authors is from this institution, and the institution which provided the data was not acknowledged at all by the authors. How did the authors acquire these data?

Re:Thank you for your comments. I have made modifications according to your comments:I refer to the literature.

  1. The conclusions of this study was obtained only based on the 2017 national urban public security survey data in the urban public security database of China University of Mining and Technology. Attention should be taken that the government governance on regional public health and development measures might have changed a lot. I do not think the results of this study could reflect the current impact of government governance on regional public health.

Re:Thank you for your comments. I have made modifications according to your comments.

  1. In the section of conclusion, the authors stated “, and also makes the father-in-law question the government's governance ability in the field of public health”. What the hell the authors mean???

Re:Thank you for your comments. I have made modifications according to your comments:However, in the new era of social transformation, various public health events emerge in an endless stream, which leads to a great threat to public health, and also makes people question the government's governance capacity in the field of public health. In this case, the government should aim at building a service-oriented government, give full play to its initiative in performing public health functions, and strive to reform and improve China's public health undertakings under the guidance of the Public Health Law.